# Improved Microstructure and Hardness Properties of Low-Temperature Microwave-Sintered Y_2_O_3_ Stabilized ZrO_2_ Ceramics with Additions of Nano TiO_2_ Powders

**DOI:** 10.3390/ma13071546

**Published:** 2020-03-27

**Authors:** Min-Hang Weng, Cheng-Xun Lin, Cian-Song Huang, Chin-Yi Tsai, Ru-Yuan Yang

**Affiliations:** 1School of Information Engineering, Putian University, Putian 351100, Fujian, China; hcwweng@gmail.com; 2Graduate Institute of Materials Engineering, National Pingtung University of Science and Technology, Pingtung County 912, Taiwan; b10238045@mail.npust.edu.tw (C.-X.L.); tylertsaiji@gmail.com (C.-Y.T.); 3Department of Food Science, National Pingtung University of Science and Technology, Pingtung County 912, Taiwan; abc340414@gmail.com

**Keywords:** zirconia, microstructure, Vickers hardness, microwave sintering

## Abstract

This paper reports the improvement of microstructural and hardness properties of 3 mol% yttria-stabilized zirconia (3Y-TZP) ceramics with nano TiO_2_ powders (with 0, 0.9, 1.8, and 2.7 wt%) added using a low-temperature microwave-assisted sintering of 1250 °C. Even at such a low sintering temperature, all sintered samples had the main phase of tetragonal zirconia (t-ZrO_2_) without the appearance of the secondary monoclinic phase or TiO_2_ phase, and had high relative densities, larger than 95%. The grain growth was well developed, and the grain sizes were around 300–600 nm. The Ti and O elements appeared at the grain and grain boundary and increased with the increased nano TiO_2_ contents identified by the element analysis, although the TiO_2_ phase did not appear in the X-ray pattern. The Vickers hardness was in the range of 10.5 to 14.5 GPa, which first increased with increasing content till 0.9 wt% and then decreased. With citric acid corrosion treatment for 10 h, the Vickers hardness only decreased from 14.34 GPa to 13.55 GPa with the addition of 0.9 wt% nano TiO_2_ powder. The experiment results showed that 0.9 wt% nano TiO_2_ addition can improve the densification as well as the Vickers hardness under a low temperature of microwave-assisted sintering.

## 1. Introduction

Zirconia (ZrO_2_) has become one of the most important dental materials in the recent development of restorative dentistry, due to the finding of transformation toughening mechanisms [1,2]. Based on the final composition and temperature used, three different polymorphs are known to exist in pure zirconia, including monoclinic zirconia (m-ZrO_2_), tetragonal zirconia (t-ZrO_2_), and cubic zirconia (c-ZrO_2_). It is well known, based on equilibrium conditions, that the monoclinic (m) phase exists stably at room temperature to 1170 °C, the tetragonal (t) phase occurs and is stable at temperatures between 1170 °C and 2370 °C, and the cubic (c) phase is stable above 2370 °C [3]. Additionally, ZrO_2_ has many special chemical, physical, and mechanical properties, for example, low wear resistance, chemical inertness, ionic conductivity, high elastic modulus, high strength, high toughness, high elastic modulus, and high melting temperature. Due to these excellent properties, ZrO_2_ is popular as an engineering material and is a replacement to alumina as a dental material [4].

Although ZrO_2_-based ceramics are very popular for dental use due to their excellent hardness, toughness, and fracture strength, they still have a drawback since the low-temperature cubic phase and tetragonal phase are metastable and might be transformed into the monoclinic phase (with a low hardness property) during the cooling process after sintering [1,2]. Keeping the tetragonal phase stable at the low-temperature state to provide the desired mechanical properties is required for increasing the biomedical applicability [1]. In the past, many techniques have been reported for holding the tetragonal phase of the ZrO_2_-based ceramics and avoiding the damaging t → m phase transformation during the cooling process [3,5]. For example, Partially Stabilized Zirconia (PSZ) material is well developed by adding a small content of stabilizers, such as Y_2_O_3_, CeO_2,_ and MgO, into ZrO_2_-based material to provide various coexistent phases that can hold the tetragonal phase from 1170°C to room temperature and prevent the appearance of crack propagation, thus giving high hardness and fracture toughness [4,5]. Furthermore, near full densification and uniform grain size of the ZrO_2_-based ceramics are also needed in order to enhance the mechanical properties.

In various PSZ materials, 3 mol% yttria-stabilized zirconia, known as 3 mol% yttria-stabilized zirconia (3Y-TZP), provides a very fine yttria grain in the zirconia polycrystals, showing only the tetragonal phase at room temperature and good mechanical properties, including high hardness, toughness, and strength [6,7,8,9]. Thus, 3Y-TZP is a desired material for the soft machining of dental prosthetic systems requiring long-term durability.

Recently, it was reported that small amounts of oxide, such as MgO, CaO, TiO_2_, HfO_2_, CeO_2_, Nb_2_O_5_, and Ta_2_O_5_, can be doped into the 3Y-TZP to further increase the phase stability by decreasing the axial ratio c/a of the structure [10]. Moreover, 7.7 mol% TiO_2_-doped 3Y-TZP has been used for the joining technique of dental bridges as an insert material, since TiO_2_-doped 3Y-TZP can provide enhanced plasticity. With the suitable mixture of particle sizes, 3Y-TZP composites can increase grain densification and improve the mechanical properties of the ceramics [11].

As is well known, furnace sintering provides heat that is radiantly transferred to the surface of the sintered sample and is spread to the material core through thermal conduction, thus causing nonuniform heating distribution. Sintering of ZrO_2_ dental samples usually requires a long sintering time from 8 to 10 h at a temperature between 1400 and 1550 °C to obtain the relative density of better than 98% in the sintered material, thus causing much energy consumption [2,3]. Unlike conventional heating with large thermal energy losses, microwave heating provides electromagnetic energy to the materials, and heat is generated internally within the material [12]. The process is very rapid, since the material is heated by energy conversion instead of energy transfer. The conversion of electromagnetic energy into heat is almost 100%, largely within the sample itself [13,14,15]. It was reported that ZrO_2_ ceramics with different amounts of micrometer TiO_2_ powder added could be microwave-sintered at 1300 °C. However, the sintered samples had many monoclinic phases instead of the tetragonal phases and thus had poor hardness of only 125 to 300 Hv [16]. It was reported that small contents of CuO powders were added to 3Y-TZP ceramics to obtain a high hardness at low microwave-sintering temperatures [17]. Therefore, it was expected that the nano TiO_2_ powder added to 3Y-TZP ceramics using microwave sintering would improve the microstructure and mechanical properties.

In this study, the purpose was to study the effect of the addition of nano TiO_2_ powders on the phase, grain size, and hardness of the 3Y-TZP at a low microwave sintering temperature of 1250 °C. Phase and grain growth were observed by X-ray diffraction (XRD) and scanning electron microscopy (SEM), respectively. Hardness was measured by the Vickers detection. Furthermore, the dependence of the nano TiO_2_ powers with 0, 0.9, 1.8, and 2.7 wt% on the hardness property was investigated to discuss the importance of nano TiO_2_ addition for the as-sintered samples and the sample with the treatment of citric acid corrosion.

## 2. Experiment

### 2.1. Sample Preparation

In this study, 3Y-TZP powders with purity of 94.3% were used as the starting powders. Nano TiO_2_ powders with purity of 99.85% and particle size of 300–600 nm were used as the doping material. 3Y-TZP powders had 0, 0.9, 1.8, and 2.7 wt% nano TiO_2_ powders added to them. The mixed powders were ZrO_2_-ball milled for 10 h in ethanol, to obtain a uniform mixing and then dried at 90 °C for 10 h. The dried powders were machine-grinded and a 5 wt% polyvinyl alcohol (PVA) solution was added. Tested samples were formed by uniaxially pressing to a 10 mm diameter and a 2 mm thickness. The samples, surrounded with four SiC plates, were microwave-assisted sintered in a microwave sintering oven. The sintered samples burned out the binder at 300 °C for 1 h and were then further heated at the sintering temperature of 1250 °C with a holding time of 1 h under an air atmosphere during the microwave-assisted sintering process. After microwave sintering, the microwave sintering oven was turned off and the samples were cooled slowly to room temperature without external heating [15,16].

### 2.2. Characterization

The diameter and thickness of the microwave-sintered samples were determined by using a digital caliper to calculate the shrinkage percent. X-ray diffraction (XRD, Bruker D8 Advance, Bruker Corporation, MA, USA) analysis with CuKα radiation of λ = 1.5406 Å using a Ni filter and with a secondary graphite monochromator was used to conduct the crystalline phases of the microwave-assisted sintered samples. A scanning range of 2θ = 20°–80° with a step of 0.03° and a count time per-step of 0.4 s was used. The scanning electron microscope (SEM, HORIBA EX-200, HORIBA, LTD., Kyoto, Japan) with the function of energy dispersive spectroscopy (EDS) was used to examine the surface morphology and the element content of the microwave-sintered samples. The linear intercept method on the surfaces of the SEM images was used to obtain the average grain sizes of the sintered samples [18]. To measure the hardness, a micro-Vickers hardness tester (HM-113, Mitutoyo Corporation, Kanagawa, Japan) was used, with an indent load of 1g held for 10 s on the surface of the microwave-sintered samples. To evaluate the effect of citric acid corrosion, the microwave-sintered samples were exposed in citric acid for 10 h. Citric acid solution was prepared by adding 0.5 g of citric acid powder to 40 mL of pure water, and then stirring was carried out with a magnetic stirrer until the powder was completely dissolved. The citric acid solution was obtained with a pH of 2.14. The microwave-sintered samples with citric acid corrosion were taken out for hardness test.

## 3. Results and Discussion

### 3.1. Microstructure

Figure 1 shows the XRD results of the microwave-sintered 3Y-TZP samples with the addition of 0, 0.9, 1.8, and 2.7 wt% TiO_2_ powders, sintered at 1250 °C for 1 h. It was clearly observed that all the diffraction results were in the tetragonal phase, having 2θ = 30°, 51°, and 60°, without any observation of a secondary monoclinic phase or TiO_2_ phase for different TiO_2_ contents [2,6]. It is known that the cubic and tetragonal phases of ZrO_2_ ceramics are metastable at room temperature, and would have been transformed into the monoclinic phase during the cooling process [5]. By adding the stabilizer, Y_2_O_3_, 3Y-TZP ceramic showed stable tetragonal structure, even at the room temperature. In reference [16], the ZrO_2_-based ceramics with micro-sized TiO_2_ additions could be microwave-sintered at 1300 °C; however, the XRD results showed the sintered samples had many monoclinic phases instead of the tetragonal phases. In this study, TiO_2_ powders at nano-size were used. Therefore, it was found that with the addition of the nano-sized TiO_2_ powders instead of the micro-size TiO_2_ powders, microwave sintering was more useful to form the stable phase of the 3Y-TZP, even with a low sintering temperature of 1250 °C, which is much lower than that used in conventional sintering [19].

Figure 2 shows the absolute density and the relative density of the microwave-sintered 3Y-TZP samples with addition of nano TiO_2_ powders. Under the sintering temperature of 1250 °C and the holding time of 1 h, the linear shrinkage of the microwave-sintered 3Y-TZP samples with different additions are about 20%–22%. In the density measurement, 3Y-TZP samples with 0, 0.9, 1.8, and 2.7 wt% addition of nano TiO_2_ powders have absolute densities of 5.65, 5.92, 5.72 and 5.45 g/cm^3^, respectively. Namely, these relative densities of microwave-sintered 3Y-TZP ceramics are around 93.4%, 97.9%, 94.5%, and 90.1%, respectively. The measured results indicate that even at such a low sintering temperature of 1250 °C, with help of addition of 0.9 wt% nano TiO_2_ powder, the measured density is relatively close to the theoretical density of 3Y-TZP, thus a big fraction of the porosity is almost closed [19].

Figure 3 shows SEM pictures of the microwave-sintered 3Y-TZP ceramics with the addition of (a) 0, (b) 0.9, (c) 1.8, and (d) 2.7 wt% nano TiO_2_ powders, sintered at 1250 °C for 1 h. It is clearly observed that the grains of the 3Y-TZP samples with the addition of nano TiO_2_ powders ultimately coalesced with each other, indicating the aggregated grains had clearly grown. Without the addition of nano TiO_2_ powders, some pores were still present at the triple junctions of the 3Y-TZP grains. The existing pores enlarged the diffusion distance between 3Y-TZP grains, and the shrinkage ability of the sintered ceramics was reduced [2]. The addition of nano TiO_2_ powders in 3Y-TZP ceramic could slightly improve the grain growth. The reason might be that the Ti ion is a transition element, which absorbs electromagnetic energy quickly.

The grain sizes were estimated as 350, 460, 520, and 560 nm for the additions of nano TiO_2_ powders with 0, 0.9, 1.8, and 2.7 wt%, respectively. It was reported that when grain size was larger than a critical grain size of 1000 nm, 3Y-TZP would reduce in stability and be more likely to spontaneously transformation from t→m, and when grain size was smaller than 1000 nm, 3Y-TZP would lower the transformation rate [20]. Therefore, the prepared samples can be stable without the phase transformation.

In this study, the TiO_2_ content was too small, thus the TiO_2_ phase was hard to detect in the XRD pattern since the detection of the XRD had a limitation of around 3% [21]. The element analysis was used to investigate the location of the nano TiO_2_ powders around the grain and/or the grain boundary. Figure 4 shows EDS analysis of the microwave-sintered 3Y-TZP ceramics as function of (a) 0, (b) 0.9, (c) 1.8, and (d) 2.7 wt% nano TiO_2_ powders. The Ti and O elements were identified on both the grain and grain boundaries. It might be because the radius of the Ti ion and Zt ion are very close, around 0.74 Å and 0.86 Å, respectively [22]. Moreover, nano TiO_2_ powder reacted more easily than micro TiO_2_ powder to the 3Y-TZP material before the limitation of solid solution. Figure 5 shows TiO_2_ contents in the grain and grain boundary of the microwave-sintered 3Y-TZP samples sintered at 1250 °C. The contents of TiO_2_ in the grain and grain boundaries both increased with increasing the addition content of the nano TiO_2_ powders.

### 3.2. Hardness Test

Hardness is used to represent a material’s resistance to plastic deformation, and is typically measured as nondestructive testing [1]. The calculated hardness values were mainly affected by the surface quality of the prepared sample, the hardness tester used, and the effects of environmental and human factors.

Moreover, the important subject in ZrO_2_-based ceramics is the sensitivity of low-temperature degradation (LTD) [23,24]. It was reported that transformation usually first begins around the isolated grains on the surface by a stress corrosion mechanism [20]. In this study, the effect of citric acid corrosion was evaluated. The microwave-sintered samples were exposed to a citric acid solution.

Figure 6 shows typical optical microscope of the 3Y-TZP ceramics with 0.9 wt% nano TiO_2_ powders added, both (a) as microwave-sintered and (b) under the treatment of citric acid corrosion for 10 h. Typically, the shorter length of the cross-indentation indicates the larger Vickers hardness. Figure 7 summarizes the Vickers hardness of the 3Y-TZP ceramics as a function of various additions of 0, 0.9, 1.8, and 2.7 wt% nano TiO_2_ powders under as-sintering and after treatment of citric acid corrosion.

The Vickers hardness of the as-sintered 3Y-TZP ceramics with the addition of 0, 0.9, 1.8, and 2.7 wt% nano TiO_2_ powders was calculated as 12.95, 14.34, 10.97, and 10.48 GPa, respectively. As compared to the previous works, the measured Vickers hardness values of this study were similar to those of the CuO-added 3Y-TZP ceramics at the same sintering temperature using microwave processing [17], but much larger than those of ZrO_2_ ceramics with the addition of micro-size TiO_2_ powders at the similar sintering temperature using microwave processing [16].

Normally, there are many factors affecting the Vickers hardness of ceramic materials [22], including (1) the intrinsic factor of ceramic deformability and (2) the extrinsic factor of microstructure features, including crystal orientation, existed phases, porosity, grain size, boundary constitution, and density [6,8]. In this study, even at the low microwave sintering temperature of 1250 °C, the main phase is the tetragonal phase without any secondary monoclinic phase or TiO_2_ phase in the sintered samples with different TiO_2_ content, as discussed in Figure 1. Moreover, the grain sizes were similar for the samples with different additions of nano TiO_2_ powders, as shown in Figure 2. The measured hardness had the same trend with the density, independent of the different additions of nano TiO_2_ powders. Thus, it is suggested that in this study the hardness of the 3Y-TZP samples was mainly influenced by the relative density [18]. The highest measured hardness value, obtained from the prepared 3Y-TZP sample with a 0.9 wt% addition of nano TiO_2_ powder, may be due to the high relative density. It is noteworthy that the 3Y-TZP ceramics with small amount of nano TiO_2_ powders also improved the densification as well as the hardness in both the as-sintered and citric-acid-treated samples.

In addition, the Vickers hardness of the microwave-sintered 3Y-TZP ceramics as a function of various additions of 0, 0.9, 1.8, and 2.7 wt% nano TiO_2_ powders after the treatment of citric acid corrosion for 10 h were calculated as 12.53, 13.55, 12.62, and 12.83 GPa, respectively.

Typically, transformation from the tetragonal to the monoclinic phase of zirconia-based ceramic is strongly accelerated in humid or solution environments, causing a distribution of structural integrity and poor hardness [20]. However, as shown in Figure 7, the measured hardness values of the citric acid treated samples only decreased from 14.34 GPa to 13.55 GPa at the addition of 0.9 wt% nano TiO_2_ powder, and even higher than those of the as-sintered samples with increased additions of nano TiO_2_ powders to 2.7 wt%.

The first reason hardness may have improved in citric acid treated samples may be attributed to size-induced phase stabilization. The phase stability of the 3Y-TZP material in hot water or solution is greatly affected by grain size, since t-m transformation is considerably enhanced as grain size increases. The high-temperature phases become more stable than the low-temperature ones when the grain size is below a critical grain size [25]. As shown in Figure 3, for all the samples with the addition of nano TiO_2_ powders of 0, 0.9, 1.8, and 2.7 wt%, the grain sizes were smaller than the critical grain size of around 1000 nm. The second reason for improved hardness in citric acid treated samples may be attributed to the high densification due to the existence of the addition of stabilizer and nano TiO_2_ powder. The microwave sintering for the 3Y-TZP samples with various additions of nano TiO_2_ powders improved the densification and reduced the porosity of the sintered samples, decreasing the nucleation of the monoclinic phase [26]. A highly dense sample, having a measured density close to the theoretical density reduced the transformation rate [3]. The treatment of citric acid corrosion provided a clean surface to the sintered samples for the samples with additions of 1.8 wt% and 2.7 wt% nano TiO_2_ powders, thus enhancing the measured hardness. Both parameters of grain size and densification resulted in an unaffected stability under the treatment of citric acid corrosion.

It was reported that the Vickers hardness values of 0.3 and 0.5 wt% MnO_2_-doped yttria tetragonal zirconia polycrystal (Y-TZP) ceramics sintered at 1250 °C had a high hardness of ~13.2 GPa (0.3 wt%) and 13.6 GPa (0.5 wt%) [27], and that the Vickers hardness values of doped Al_2_O_3_ and CeO_2_-doped Y-TZP ceramics sintered at 1400 °C were around 14 GPa [28]. As compared to these reports, the maximum Vickers hardness value of 14.34 GPa for the as-sintered TiO_2_-doped 3Y-TZP sample at 1250 °C was relatively excellent, and even the Vickers hardness value of 13.55 GPa for the citric acid treated sample was also high enough.

From the above results, it is summarized that microwave sintering with help of additions of nano TiO_2_ powder showed an important factor on the phases, grain growth, and hardness of 3Y-TZP ceramics sintered at a low-temperature. Despite the fact their were microwave-sintered and citric-acid-treated samples, the measured hardness values were high enough to be designed for dental application.

## 4. Conclusions

In this paper, the improvement of microstructural and hardness properties of 3Y-TZP bio-ceramics with the addition of nano TiO_2_ powder is investigated. The 3Y-TZP ceramics with additions of 0, 0.9, 1.8, and 2.7 wt% nano TiO_2_ powders were sintered at a low sintering temperature of 1250 °C. For all the sintered samples, the phase was the tetragonal zirconia (t-ZrO_2_) phase without monoclinic zirconia (m-ZrO_2_) phase and secondary phase of TiO_2_, and the aggregated grains had clearly grown. The estimated grain sizes were around 300-600nm without and with the addition of nano TiO_2_ powders. The Vickers hardness had a value from 10.5 to 14.5 GPa, and increased first with the increase of TiO_2_ content until 0.9 wt% and then decreased. Compared to the as-sintered samples, the Vickers hardness of the citric acid treated samples was only decreased from 14.34 GPa to 13.55 GPa for samples with the addition of 0.9 wt% nano TiO_2_ powders. The experiment results showed that addition of 0.9 wt% nano TiO_2_ powders improved the densification and the Vickers hardness of the 3Y-TZP samples, even microwave-sintered at a relatively low temperature.

## Figures and Tables

**Figure 1 materials-13-01546-f001:**
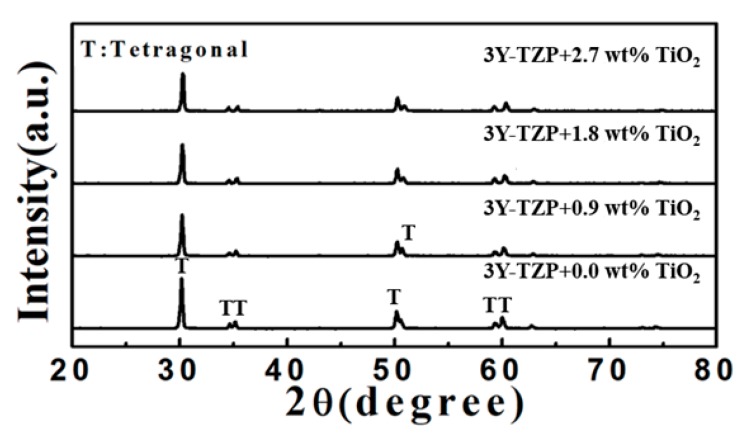
XRD patterns of the microwave-sintered 3 mol% yttria-stabilized zirconia (3Y-TZP) samples with the addition of 0, 0.9, 1.8, and 2.7 wt% nano TiO_2_ powders.

**Figure 2 materials-13-01546-f002:**
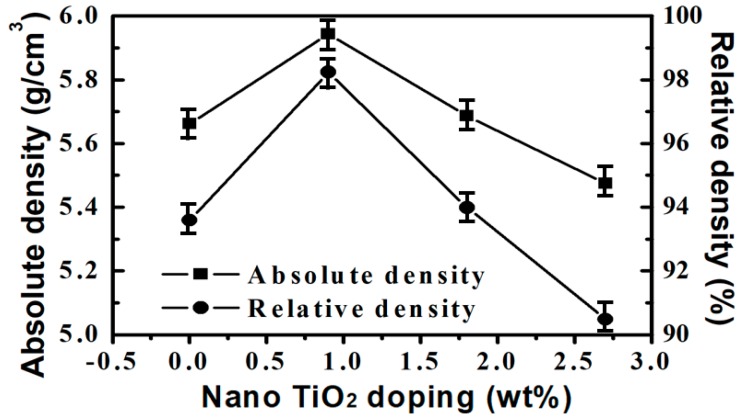
Absolute density and relative density of the microwave-assisted sintered 3Y-TZP samples as function of 0, 0.9, 1.8, and 2.7 wt% nano TiO_2_ contents microwave-assisted sintered at 1250 °C.

**Figure 3 materials-13-01546-f003:**
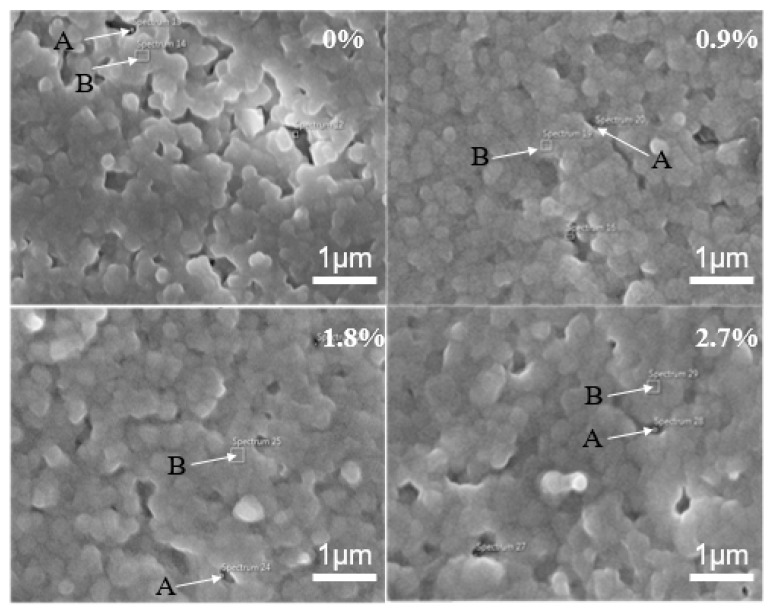
SEM pictures of the microwave-sintered 3Y-TZP ceramics as function of nano TiO_2_ powders, sintered at 1250 °C (A is directed to grain and B is directed to grain boundary).

**Figure 4 materials-13-01546-f004:**
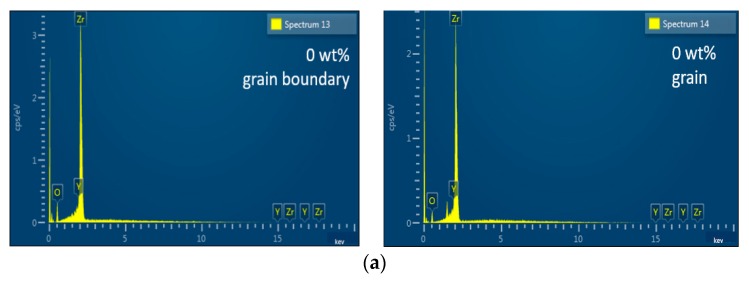
EDS analysis of the microwave-sintered 3Y-TZP ceramics as function of (**a**) 0, (**b**) 0.9, (**c**) 1.8, and (**d**) 2.7 wt% nano TiO_2_ powders, sintered at 1250 °C.

**Figure 5 materials-13-01546-f005:**
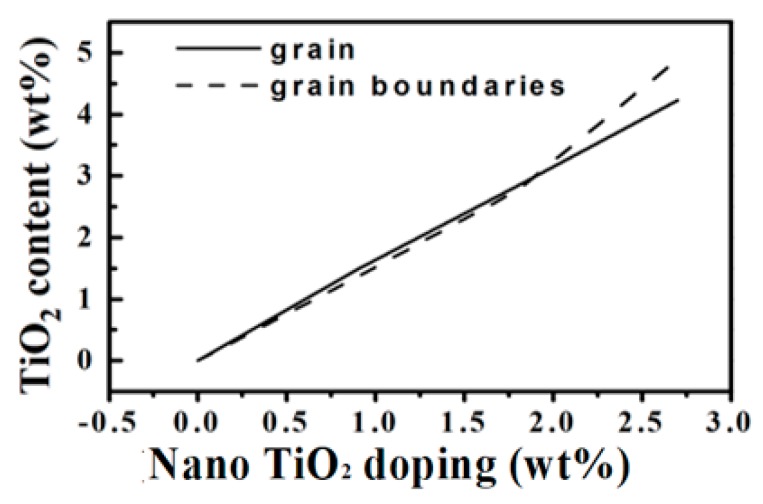
TiO_2_ contents in grains and grain boundaries of the microwave-sintered 3Y-TZP ceramics sintered at 1250 °C.

**Figure 6 materials-13-01546-f006:**
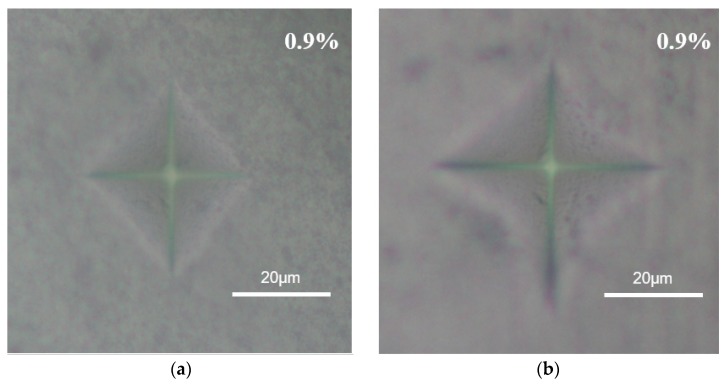
Optical microscope of the 3Y-TZP ceramics added with 0.9 wt% nano TiO_2_ powders. (**a**) as microwave-sintered and (**b**) under the treatment of citric acid corrosion for 10 h.

**Figure 7 materials-13-01546-f007:**
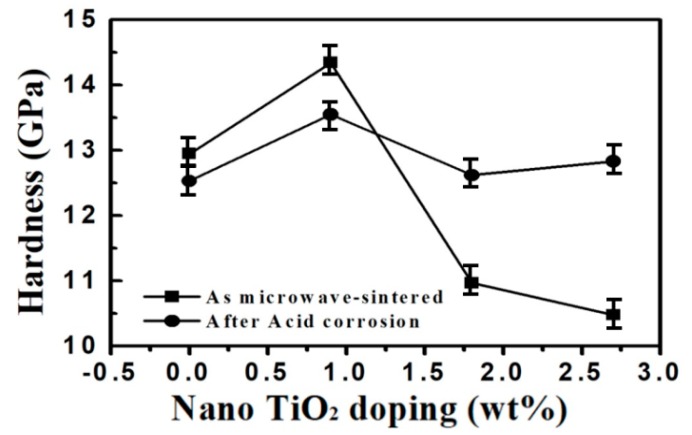
Vickers hardness of the 3Y-TZP ceramics as a function of 0, 0.9, 1.8, and 2.7 wt% nano TiO_2_ contents under as-sintering and after citric acid corrosion treatment.

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
