# Peer review of "Improved Microstructure and Hardness Properties of Low-Temperature Microwave-Sintered Y2O3 Stabilized ZrO2 Ceramics with Additions of Nano TiO2 Powders"

_materials, 2020, doi:10.3390/ma13071546_

Round 1
Reviewer 1 Report
Please see attached file

Reviewer 2 Report
Title: Improved Microstructure And Mechanical Properties 1 of Low Temperature Microwave-Sintered Y2O3 2 Stabilized ZrO2 Ceramics With Additions of Nano 3 TiO2 Powders
Authors: Min-Hang Weng, Cheng-Xun Lin, Cian-Song Huang, Chin-Yi Tsai and Ru-Yuan Yang
General comments: In this study the microstructural (phase, grain size and boundaries) and hardness of doped nano TiO2 - 3Y-TZP is investigated.
Abstract: Mentioning “without appearance of the second phase”, there are several phases of stabilized zirconia. Which one do you refer to? This are also mentioned in Results and discussion, please add some information for additional explanation.
“The Ti and O elements are appeared at the grain and grain boundary and increased with the increased nano TiO2 contents identified by the element analysis, although the TiO2 phase was not appeared in the X-ray pattern.” How can Ti and O be identified, if the TiO2 pahse was not appeared in the X-ray pattern?
“The Vickers hardness is not affected too much.” and in Results and Discussion: “hardness values of the citric acid treated samples are not decreased too much, and even higher than those of the as-sintered samples as increasing of the additions of nano TiO2 powders to 2.7 wt%. “too much” – what is the definition of too much?
Stating that the material is “appropriate for dental applications”, based only on densification and hardness test, consider to rephrase the conclusion.
Aim: The aim include the effect of adding TiO2 with different amount of wt% and with low microwave sintering temp. on following: phase, grain size, (evaluated by XRD and SEM) hardness ( by Vickers) and investigate the mechanical properties – including corrosion - how is this (more than hardness test) evaluated and highlighted in the paper?
Material and methods:Sample preparation; please define PVA solution
How does uniaxially pressing and the chosen sample size (10 mm diameter and a 2 mm thickness) impact/effect the results?
“The sintered samples were burned out the binder at 300°C for 1h” – was this during the microwave sintering process or after sintering, the samples were heat-treated?
After sintering, cooled naturally down to RT - please define naturally?
Results and discussion: What does “would be transformed into the monoclinic phase for a short time (line 124)” actually mean? The phase transformation from t- to m-phase will not be reversible without any heat annealing.
Conclusion: “In this paper, the improvement of microstructural and mechanical properties of 3Y-TZP bio- ceramics with addition of nano TiO2 powder is investigated.” This is general, the study is evaluating in a more limited way than general mentioned.
“and the grain growth is almost finished” What does that actually mean and what effect does that have on the doped zirconia?
“the Vickers hardness is not affected too much” As previously mentioned, please define too much?
“which makes it suitable for specific dental applications and reduces the energy consumption.” Can this actually be concluded, have you evaluated the energy consumption?
Figure 1. Cannot be review, the figure is missing in the document
Figure 2. Cannot be review, the figure is missing in the document
Figure 3. Cannot be review, the figure is missing in the document
Figure 8. Cannot be review, the figure is missing in the document
Reviewer 3 Report
Several figures and forms are incorrectly mentioned like dilate. Therefore, it should be sent back to authors, then, correct format.
Round 2
Reviewer 1 Report
All comments and suggestions have been addressed adequately.
Author Response
We would like thank Reviewer's work and suggestions.Reviewer 3 Report
In figure1, please add zirconia legends.
In figure4, please add legend of x-axis.
Author Response
We would like thank Reviewer's work and suggestions, and we have amended the Figure 1 and Figure 3 to add zirconia (3Y-TZP) legends in figure 1 and legends of x-axis in figure 4.